# Nomograms for Predicting Disease-Free Survival Based on Core Needle Biopsy and Surgical Specimens in Female Breast Cancer Patients with Non-Pathological Complete Response to Neoadjuvant Chemotherapy

**DOI:** 10.3390/jpm13020249

**Published:** 2023-01-29

**Authors:** Ailin Lan, Han Li, Junru Chen, Meiying Shen, Yudi Jin, Yuran Dai, Linshan Jiang, Xin Dai, Yang Peng, Shengchun Liu

**Affiliations:** 1Department of Breast and Thyroid Surgery, The First Affiliated Hospital of Chongqing Medical University, No. 1 Youyi Road, Yuzhong District, Chongqing 400016, China; 2Department of Cardiothoracic Surgery, The First Affiliated Hospital of Chongqing Medical University, No. 1 Youyi Road, Yuzhong District, Chongqing 400016, China; 3Department of Pathology, Chongqing University Cancer Hospital, No. 181 Hanyu Road, Shapingba District, Chongqing 400030, China

**Keywords:** breast cancer, neoadjuvant chemotherapy, prognosis, nomogram, pathological complete response

## Abstract

Purpose: While a pathologic complete response (pCR) is regarded as a surrogate endpoint for pos-itive outcomes in breast cancer (BC) patients receiving neoadjuvant chemotherapy (NAC), fore-casting the prognosis of non-pCR patients is still an open issue. This study aimed to create and evaluate nomogram models for estimating the likelihood of disease-free survival (DFS) for non-pCR patients. Methods: A retrospective analysis of 607 non-pCR BC patients was conducted (2012–2018). After converting continuous variables to categorical variables, variables entering the model were progressively identified by univariate and multivariate Cox regression analyses, and then pre-NAC and post-NAC nomogram models were developed. Regarding their discrimination, ac-curacy, and clinical value, the performance of the models was evaluated by internal and external validation. Two risk assessments were performed for each patient based on two models; patients were separated into different risk groups based on the calculated cut-off values for each model, including low-risk (assessed by the pre-NAC model) to low-risk (assessed by the post-NAC model), high-risk to low-risk, low-risk to high-risk, and high-risk to high-risk groups. The DFS of different groups was assessed using the Kaplan–Meier method. Results: Both pre-NAC and post-NAC nomogram models were built with clinical nodal (cN) status and estrogen receptor (ER), Ki67, and p53 status (all *p* < 0.05), showing good discrimination and calibration in both internal and external validation. We also assessed the performance of the two models in four subtypes, with the tri-ple-negative subtype showing the best prediction. Patients in the high-risk to high-risk subgroup have significantly poorer survival rates (*p* < 0.0001). Conclusion: Two robust and effective nomo-grams were developed to personalize the prediction of DFS in non-pCR BC patients treated with NAC.

## 1. Introduction

Breast cancer (BC) is the most common female malignant tumor and the fifth leading cause of cancer mortality after lung cancer, colorectum cancer, liver cancer, and stomach cancer [1]. According to current international guidelines, BC is classified into several subtypes based on the molecular profiling of the estrogen receptor (ER), the progesterone receptor (PR), and human epidermal growth factor receptor 2 (HER2), including hormone receptor-positive (HR-positive)/HER2-negative, HR-positive/HER2-positive, HR-negative/HER2-positive, and triple-negative breast cancer (TNBC) [2]. Different subtypes with distinct clinical and pathological characteristics result in statistically significant differences in survival outcomes [3,4].

Neoadjuvant chemotherapy (NAC), also known as preoperative chemotherapy, is an effective treatment widely used before surgery and is the current standard of care for locally advanced or high-risk early-stage BC [5], with the goal of downstaging the tumor and rendering inoperable tumors operable [6]. NAC is recommended for individuals with locally advanced or inoperable breast cancer, particularly those with N2 and N3 regional lymph node nodal disease and T4 tumors, according to the NCCN Guidelines Version 3.2022 [2]. If the patient’s breast cancer subtype is linked to a high likelihood of a response, NAC may also be administered to patients with operable tumors [2]. Furthermore, crucial prognostic information can be obtained from the response to NAC [7].

After the completion of NAC treatment, the pathological response of tumors to NAC will be assessed. A pathologic complete response (pCR) is considered to be ypT0/Tis ypN0, meaning there are no residual malignant cells in any excision of breast tissues or lymph nodes after the completion of NAC treatment [8]. In patients with BC, the connection between pathologic response and long-term survival outcomes is highest in patients with TNBC, less so in those with HER2-positive cancer, and lowest in patients with HR-positive cancer [7]. The pCR is considered a reliable and irreplaceable predictor of favorable survival outcomes after NAC treatment for all BC molecular subtypes [9], while how to predict survival outcomes in non-pCR patients is an ongoing question.

p53 is a common immunohistochemical biomarker in the pathological examination of BC. The main reason for the trendy research on p53 is that abnormalities in the p53 signaling pathway are present in almost all human tumors, and mutations in p53 are present in nearly 50% of malignant tumors [10]. In general, 30–35% of invasive primary breast malignancies have a p53 gene mutation. However, in breast cancer, the prevalence of p53 mutations varies depending on the subtypes of the disease, being present in roughly 80% of TNBC, 10% of luminal A disease, 30% of luminal B disease, and 70% of HER2-enriched disease [11,12,13]. However, the role of p53 in breast cancer is often overlooked compared to HR and HER2, probably because no p53-based therapies have been approved to date. According to a high-impact review that was just published, the idea that there is no treatment for p53 is being increasingly debunked [14]. If effective, it has the potential to change cancer treatment due to the importance of p53 in human cancer. Although p53 is not currently a standard parameter in breast cancer and is not included in international guidelines, existing studies have found that p53 mutations are present in all molecular subtypes of BC [11] and are associated with poorer survival outcomes [15]. However, contrary to that, studies on the efficacy of NAC for BC have shown that pCR rates are significantly higher in tumors with p53 mutations compared to wild-type tumors [16].

Because the nomogram can reduce complex factors into a simple numerical estimation model to predict the probability of an event, it is widely used in cancer prediction modeling. A considerable number of nomogram prediction models have been developed [17]; however, the biomarkers involved in most of them are not available for every patient due to cumbersome steps or economic reasons. The purpose of this study was to fill this research gap by developing and evaluating novel nomograms for predicting disease-free survival (DFS) based on core needle biopsy and surgical specimens in female BC patients with a non-pCR to NAC. A biopsy at diagnosis and post-NAC residual tumors in patients without a pCR can show discrepancies in receptor status (from negative to positive or vice versa) and changes in biomarker expression [18,19]. Discordances in ER, PR, HER2, Ki67, and p53 can indicate chemosensitivity or chemoresistance as well as have clinical implications for the prognosis and adjuvant therapy [18,20,21,22]. Therefore, changes in biomarkers after NAC were considered in this study. To our knowledge, this is the first survival prediction model study that considers changes in biomarkers after NAC and involves p53 expression levels in BC patients treated with NAC.

## 2. Materials and Methods

### 2.1. Study Subjects

We conducted a retrospective analysis of female patients with invasive BC confirmed by core biopsy and treated with NAC without achieving a pCR at the First Affiliated Hospital of Chongqing Medical University between January 2012 and December 2017. Participants who met the following inclusion criteria were included: (1) female patients ≥18 years of age; (2) pathologically confirmed BC; (3) patients who underwent NAC and subsequent surgery; (4) complete follow-up information; (5) no distant metastasis before NAC; (6) no history of contralateral BC; (7) no synchronous bilateral BC. Patients were excluded if they (1) achieved a pCR after NAC; (2) received a non-anthracycline- and taxane-based NAC regimen; (3) lacked specific values for pre- and post-NAC ER, PR, Ki67, and p53; (4) had unknown HER2 status. Finally, a total of 476 patients were included in the primary cohort (Figure 1). According to the same standard, patients who received NAC between January 2018 and December 2018 were gathered as the validation cohort, which would serve as the nomogram’s validation group. A total of 131 patients were included in the validation cohort. This study was approved by the Ethics Committee of the First Hospital of Chongqing Medical University (ID: No. 2020–59). All procedures in the study were performed following the standards of the Institutional Research Board and the Declaration of Helsinki. Patients’ personal information was protected and is not shown.

NAC was administered following the regional protocol and national guidelines. Every three weeks, TEC NAC regimens (docetaxel 75 mg/m^2^, epirubicin 75 mg/m^2^, and cyclophosphamide 500 mg/m^2^) were given. Within a week of diagnosis, all patients began their first NAC cycle. Trastuzumab was added to the regimen in the case of HER2(+) disease; trastuzumab was administered intravenously once every three weeks at a loading dose of 8 mg/kg, followed by 6 mg/kg. However, only an exceedingly tiny percentage of HER2-positive patients received neoadjuvant targeted therapy due to health insurance and financial considerations.

### 2.2. Pathologic Assessment

All core needle biopsy specimens and surgical specimens were fixed in formalin solution within 2 h after separation and then sent to the pathology department of Chongqing Medical University. All specimens were processed in the pathology department according to immunohistochemistry-related procedures with the same criteria. The following are the steps involved in the immunohistochemistry (IHC) result interpretation: five high-power fields were randomly observed in the tumor’s “hottest spot”, and 100 tumor cells were analyzed in each field. Then, the average percentages (varying from 0 to 100%) of ER-, PR-, Ki67-, and p53-positive cells in the five high-power fields for each molecular marker were determined. This approach was assessed separately by two experienced pathologists, and the findings were regarded as consistent if there was a 10% or less difference in the counts produced by the two observers. If not, an agreement was reached after a reevaluation of the data (unblinded). By averaging the positive percentages indicated by the two observers, the results of the immunohistochemical interpretation were calculated. HER2 immunohistochemistry was reported as 0, 1+, 2+, or 3+ according to Sauter et al.’s [23] standards; 2+ cases were evaluated using fluorescence in situ hybridization (FISH) probes to determine the HER2 gene’s amplification status. Since the development of HER2-low as a legitimate subtype of breast cancer is accelerating, we categorized HER2 status as HER2-0, HER2-low, and HER2-positive, which is consistent with the current high-impact clinical research [24]. HER2-0 was defined as IHC 0; HER2-positive was defined as 3+ or IHC 2+ and HER2 gene amplification by FISH (HER2 gene: chromosome 17 ratio of >2.2); HER2-low was defined as IHC 1+ or IHC 2+ and the lack of HER2 gene amplification by FISH. According to the 2011 St. Gallen consensus [25], traditional HER2-negative includes HER2-0 and HER2-low. A distinction is made between ER, PR, HER2, Ki67, and p53 status before and after NAC. According to ASCO/CAP criteria [26], nuclear staining with less than 1% positive tumor cells was classified as ER/PR-negative, whereas nuclear staining with ≥1% positive tumor cells was categorized as ER/PR-positive. HR positivity is also referred to as ER and/or PR positivity. The primary tumors were divided into four subtypes according to the receptor status (positive receptor status before and/or after NAC is considered receptor positivity): HR(+)/HER2(−), HR(+)/HER2(+), HR(−)/HER2(+), and HR(−)/HER2(−). According to the Miller–Payne scoring system, a non-pCR is defined as having residual malignant cells in the excised breast tissue or lymph nodes after the completion of NAC treatment [8].

### 2.3. Clinical Lymph Node Status Assessment

Clinical lymph node status was evaluated by imaging (ultrasonography and magnetic resonance imaging).

### 2.4. Follow-Up

In the primary cohort, patients were followed up until death or 1 February 2021, and the median duration of follow-up was 60 months (95% CI 56.3–63.7). In the validation cohort, patients were followed up until death or 20 January 2023, and the median duration of follow-up was 53 months (95% CI 52.1–53.9). The survival endpoint was DFS; DFS events were defined as the first recurrence of the disease locally, regionally, or distantly, a diagnosis of contralateral BC or secondary malignancy, or death from any cause. All patients were followed up by the hospital outpatient department and by telephone.

### 2.5. Statistical Analyses

The ideal cut-off for the impact on DFS in BC patients was determined using receiver operating characteristic (ROC) curves, which enable the evaluation of the division points of a continuous variable. ROC analyses were performed to measure the critical values of ER, PR, Ki67, and p53 indicators before and after NAC, converting continuous variables to categorical variables. The Kolmogorov–Smirnov goodness-of-fit test was used to assess the normality of continuous variables, and those conforming to non-normal distributions were described as median and interquartile range and assessed using the Mann–Whitney U test; categorical variables were described as frequencies and percentages and evaluated by the chi-square test. In the primary cohort, those variables with *p* < 0.05 in univariate analysis were included in multivariate Cox regression with a forward LR method to identify independent prognostic factors affecting DFS. The “survival” R package was utilized for Cox regression analysis, and then forest plots were mapped using the “survminer” R package. The variables in the Cox regression equation were used as the final predictors to construct nomograms that predict the likelihood of 1-, 3-, and 5-year DFS. The “rms” R package was used to construct nomograms.

The concordance index (C-index) reflects the accuracy of the prognostic model; the C-index value ranged from 0.5 to 1.0, where 0.5 denotes random chance and 1.0 denotes that the model was completely capable of accurately predicting the event. The discrimination and calibration of models were validated internally in the primary cohort by the C-index, time-dependent ROC curves (using “timeROC” R package) with the area under the curve (AUC), and calibration curves using the Bootstrap method; calibration curves were plotted after 1000 repetitions of sampling of the original data using the “rms” R package. External validation was performed in the validation cohort using time-dependent ROC curves and calibration curves. Decision curve analysis (DCA) performed using the “rmda “R package assessed the clinical utility and predictive value of the constructed nomograms by quantifying the net improvement benefit at different threshold probabilities.

The model-based risk score for each patient was calculated using the “survival” R package. The risk score distinguishing DFS-event tumors from DFS-event-free tumors was estimated using the “survminer” R package using Maximally Selected Rank Statistics, from which cutpoint values were obtained. Based on the model-based risk score cutpoint values, patients were divided into high-risk and low-risk groups. DFS was assessed using the “survminer” R package using the Kaplan–Meier method, and the log-rank test was used to compare differences between groups. All statistical analyses and plots were performed using IBM SPSS 26.0 (Chicago, IL, USA) software and RStudio 1.4.1 software. *p*-values < 0.05 were considered statistically significant. All methods were performed following relevant regulations and guidelines.

## 3. Results

### 3.1. Conversion of Continuous Variables to Categorical Variables

In the primary cohort, continuous variables were converted to categorical variables using ROC curve analysis to calculate the best cut-off values for ER, PR, Ki67, and p53 before and after NAC treatment to distinguish DFS-event tumors from those without DFS events. Our results showed that the optimal cut-off values for ER, PR, Ki67, and p53 before NAC treatment were 45.0% (95% CI: 0.538–0.647), 1.0% (95% CI: 0.522–0.631), 22.5% (95% CI: 0.505–0.616), and 17.5% (95% CI: 0.550–0.660), respectively (all *p* < 0.05); the best cut-off values for ER, PR, Ki67, and p53 after NAC treatment were 32.5% (95% CI: 0.566–0.673), 7.5% (95% CI: 0.507–0.615), 19.0% (95% CI: 0.521–0.633), and 17.5% (95% CI: 0.560–0.671), respectively (all *p* < 0.05). All of these results are detailed in Figure 2. Next, continuous variables were transformed into categorical variables according to their cut-off values.

### 3.2. Baseline Patient Characteristics in the Primary Cohort and Univariate Analysis

A total of 476 eligible patients in the primary cohort participated in the analysis. Of these, 147 patients (30.9%, 147/476) had a DFS event, while 329 patients (69.1%, 329/476) did not have a DFS event. The univariate analysis of predictors in the event and non-event groups showed statistically significant differences in six factors: clinical T (cT) staging, clinical nodal (cN) status, ER status, PR status, Ki67 status, and p53 status (all *p* < 0.05; ER status, PR status, Ki67 status, and p53 status included pre-NAC and post-NAC). Conversely, age at diagnosis, menopausal status, and chemotherapy cycles did not show statistically significant differences. See Table 1 for details.

### 3.3. Independent Prognostic Factors for DFS

Multivariate Cox proportional hazards regression analyses were performed for the six statistically significant predictors derived from the primary cohort described above (ER, PR, Ki67, and p53 were included in pre-NAC multivariate and post-NAC multivariate analyses based on their pre- and post-treatment status, respectively). The results of the multivariate Cox proportional hazards regression analysis are shown in a forest plot (Figure 3); only the variables that were eventually included in the equation are shown in the figure. cN, ER, Ki67, and p53 were all independent prognostic factors for DFS (either pre- or post-NAC ER, Ki67, and p53).

## 4. Nomogram Development and Validation

Nomograms were created for pre-NAC and post-NAC based on independent prognostic factors for DFS derived from a Cox proportional hazards regression analysis of the primary cohort described above, as shown in Figure 4. The discrimination of models was measured using the C-index. The C-indices of the pre-NAC and post-NAC nomograms were 0.693 and 0.701, respectively. As shown in Figure 5 for the ROC curves of the nomogram prediction model in the primary cohort, the 1-year, 3-year, and 5-year AUCs for the pre-NAC nomogram are 0.752, 0.718, and 0.702, respectively, while those for the post-NAC nomogram are 0.747, 0.737, and 0.701, respectively. We also divided the enrolled patients in the primary cohort into four subgroups based on their HR and HER2 status, namely, HR(+)/HER2(−) (206 patients), HR(+)/HER2(+) (97 patients), HR(−)/HER2(+) (96 patients), and HR(−)/HER2(−) (77 patients). Of these subgroups, HR(−)/HER2(−) had the highest performance with the maximum AUC in both pre-NAC and post-NAC nomograms. The ROC curves and associated AUCs for each subgroup are displayed in Figure 6.

Internal validation of the nomogram model in the primary cohort was performed using the Bootstrap method, and calibration curves were plotted after running 1000 replicate samples of the original data. Calibration plots of the pre-NAC and post-NAC nomograms showed that the predicted 1-, 3-, and 5-year DFS probabilities were almost identical to the actual observations, as shown in Figure 7. The DCA also shows the value of both models. The DCA’s horizontal coordinate is the threshold probability, and its vertical coordinate is the net benefit rate after deducting the disadvantage. The closer the curve is to the upper right corner, the larger the net clinical benefit predicted by this model. As shown in Figure 8, the net benefit of our prognostic model is greater than in the other two scenarios (all screening or no screening) for a wide range of threshold probabilities.

Additionally, we conducted external validation; Appendix A displays the clinical characteristics of the primary and validation cohorts. Since the validation cohort included patients who received NAC in 2018 and follow-up was not yet more than 5 years, we only validated nomograms predicting the 1- and 3-year DFS. Two models were well discriminated; the 1-year and 3-year AUCs for the pre-NAC nomogram are 0.733 and 0.773, respectively (Figure 9A), and the AUCs for the post-NAC nomogram are 0.743 and 0.773, respectively (Figure 9B). Additionally, the calibration plots in the validation cohort demonstrated good agreement (Figure 9C–F).

### Risk Stratification Based on Nomogram

Based on the curated multivariate Cox model, the risk score of each patient was calculated, the optimal cutpoint of the risk score was determined, and the total nomogram score (Total Points in Figure 4) of the patient corresponding to this optimal cutpoint of the risk score was the corresponding cutpoint value of the nomogram prognostic score. Based on the cut-off value of the nomogram prognostic score (Total Points in Figure 4), patients were classified into different risk subgroups, namely, low risk (>59.0 in the pre-NAC nomogram and >58.1 in the post-NAC nomogram) and high risk (≤59.0 in the pre-NAC nomogram and ≤58.1 in the post-NAC nomogram) (Appendix A). In the pre-NAC and post-NAC nomogram models, patients in the low-risk subgroup showed better DFS compared with the high-risk subgroup (*p* < 0.0001 for both; Appendix A). We studied the proportion of patients with changes in risk after NAC and found that the proportions of patients in low-risk to low-risk, high-risk to low-risk, low-risk to high-risk, and high-risk to high-risk groups were 71.0% (338/476), 4.0% (19/476), 5.7% (27/476), and 19.3% (92/476), respectively (Appendix A). We further determined the DFS of the above four groups. Patients in the high-risk to high-risk subgroup have significantly poorer survival rates (*p* < 0.0001; Figure 10).

## 5. Discussion

In this study, using core needle biopsy and surgical specimens, we developed nomograms for the DFS of non-pCR breast cancer. The pre-NAC and post-NAC models are both highly discriminative and have accurate calibrations in both internal and external validation. Considering the differences in biomarker expression between different breast cancer subtypes [11,27], we also assessed the performance of two models in four subtypes, with the HR(−)/HER2(−) subtype showing the best prediction. Additionally, we established the threshold for the nomogram prognostic score, allowing patients to be divided into distinct risk groups with pronounced differences in DFS between these groups. Notably, we also considered changes in biomarkers before and after NAC (which were not considered in previous studies of NAC prediction models) and classified patients into different risk groups based on the calculated cut-off values for the pre-NAC and post-NAC models, i.e., low-risk to low-risk, high-risk to low-risk, low-risk to high-risk, and high-risk to high-risk groups. We further confirm that patients in the high-risk to high-risk subgroup have significantly poorer survival rates.

Because a pCR is considered a surrogate endpoint for favorable survival outcomes in BC patients receiving NAC, many previous studies have focused on the prediction of pCR [28,29], while few studies have explored the prediction of survival in patients who have not achieved a pCR. To fill this research gap, we used patients who received NAC but did not achieve a pCR as the target population and explored the factors affecting their survival prognosis based on specimens before and after NAC treatment.

Here, we discovered that the clinical N stage, ER, Ki67, and p53 were linked to patient DFS results (either pre- or post-NAC ER, Ki67, and p53), which is consistent with previous reports showing that ER expression is a favorable prognostic factor, and lymph node positivity and increased Ki67 and p53 markers have been linked to a worse prognosis [15,30,31,32,33,34], but previous studies lacked exploration in the context of NAC for breast cancer conditions. In this condition, the impact of changes in biomarkers such as ER levels during chemotherapy also needs to be considered.

Alterations in ER, PR, HER2, Ki67, and p53 have been reported following NAC treatment, with widely varying results from series to series [18,19,21,35,36]. Considering that these biomarkers may change after NAC, we included ER, PR, HER2, Ki67, and p53 both before and after treatment in the pre-NAC and post-NAC multivariate analysis, respectively. Interestingly, we found that ER, Ki67, and p53 were significant independent predictors of DFS both before and after NAC treatment. To our knowledge, this is the first study to explore predictors of DFS outcomes in breast cancer that take into account changes in biomarkers during NAC and are based on a combination of classical parameters and immunohistochemical markers.

A recently published study found that Ki67, ER, PR, and age were independent predictors of breast-cancer-specific mortality, but p53 was not [30]. The reason for the inconsistency with our study may be that the study population was limited to ER/PR+ and HER2- breast cancer patients not receiving NAC, and p53 was simply classified as negative and positive. Additionally, in endometrial cancer, another hormone-dependent malignancy in women, ER and p53 were found to be independent predictors of recurrence in patients after surgery [37], which is in agreement with our study.

By analyzing ER, PR, Ki67, and p53 as continuous variables, we were able to classify tumors into high- and low-expression diseases. Notably, the thresholds for ER, PR, Ki67, and p53 were determined based on ROC analysis rather than simply classifying them empirically as positive and negative; this differs from previous studies [38,39].

ER is a crucial factor in defining tumor subtypes and has been widely identified as a feature that is strongly associated with breast cancer prognosis [32]; ER expression is a favorable prognostic factor and a predictor of the endocrine therapy response [40]. Our findings show that the critical values of ER before and after NAC were 45.0% and 32.5%, respectively, suggesting that higher ER expression levels before and after NAC are a crucial favorable predictor of patient prognosis. This may be explained by the similar biological behavior of ER-low and ER-negative diseases. Fujii et al. [41] found that a pre-NAC ER threshold of 10% is a better cut-off for distinguishing ER-positive from ER-negative breast cancer than 1%, but this threshold is based on speculation only and has not been statistically analyzed. In our study, PR was not an independent predictor of patient DFS, which is inconsistent with previous studies [30,42]. We speculate that this situation may be due to the interaction between ER and PR and the different study populations included. Our prognostic model identified 22.5% and 19.0% as thresholds for Ki67 status before and after NAC, respectively, which is consistent with other studies that have observed substantial observer/laboratory variation in the >5 to <30% range, which is the cut-off point chosen by most investigators [5,43].

Mutations in the TP53 gene, which encodes p53, are strongly associated with aggressive histological features and poor survival in breast cancer [44,45]. We found that the critical value of p53 was 17.5% both before and after NAC. A 10% threshold for p53 was discovered by Lee et al. to be a predictor of survival outcomes [46], which is similar to our analysis. A few studies have revealed thresholds for p53 in breast cancer; however, there are no studies on the p53 threshold in the NAC context. To our knowledge, this study is the first to investigate the function of p53 in a model used to predict NAC survival, and it also identified p53’s key predictive value. In addition, we found that pre-treatment clinical lymph node status was an independent predictor of patient survival prognosis, which is consistent with previous studies [28].

Our nomogram models can then be used to perform a risk assessment for each patient. Using a pre-NAC nomogram score of 59.0 and a post-NAC nomogram score of 58.1 as cut-offs, we logically and creatively propose dividing patients into low-risk to low-risk, high-risk to low-risk, low-risk to high-risk, and high-risk to high-risk categories. This change in risk before and after NAC is due to alterations in ER, Ki67, and p53 expression that may reflect chemotherapy sensitivity or chemoresistance and thus have clinical implications for adjuvant therapy and outcomes. We further confirm that patients in the high-risk to high-risk subgroup have significantly poorer survival rates. From this vantage point, doctors can create treatment strategies based on various risk scores. A patient should receive top-priority care and have all available treatment options explored if their pre- and post-NAC risk scores put them in the high-risk to high-risk subgroup. A small number of patients (9.7%) will have a change in risk classification after NAC, and for this group of patients, further evaluation of their risk may be required.

Although there are benefits, there are still certain restrictions. Due to the lack of histological grading data for many patients, histological grading was not included in this analysis; nevertheless, prior research has demonstrated that TP53/p53 mutations are closely related to more severe histological characteristics [44]. Additionally, there was a paucity of specific information regarding postoperative adjuvant therapy, yet systemic treatment was carried out at the time following international guidelines. By adding more prognostic risk factors, our nomogram should become better. External validation was performed at the same center, and patients in the validation cohort were not followed up for 5 years; it is envisaged that patient follow-up will continue with the performance of multicenter validation in future studies. Clinical practice would also be facilitated by a greater sample size and distinct prediction models for each subtype.

## 6. Conclusions

We have successfully created nomograms that offer a reliable and effective way of forecasting the DFS prognosis based on core needle biopsy and surgical specimens for patients who received NAC but did not achieve a pCR. Four variables, namely, cN, ER, Ki67, and p53, were included in the pre- and post-NAC models created for this investigation. These factors provided a good predictive potential for the results of DFS. Patients with varying levels of risk could be efficiently separated on this basis.

## Figures and Tables

**Figure 1 jpm-13-00249-f001:**
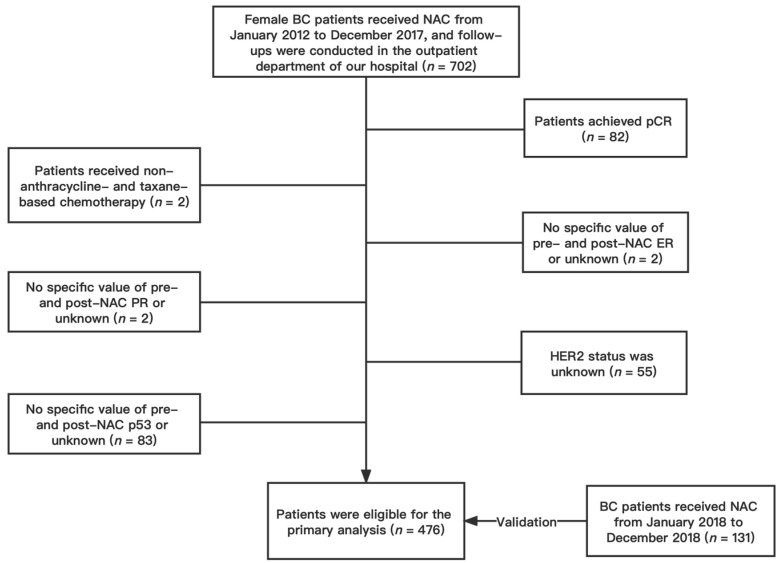
Flow chart of patient selection. Abbreviations: BC, breast cancer; NAC, neoadjuvant chemotherapy; pCR, pathologic complete response; ER estrogen receptor; PR progesterone receptor; HER2 human epidermal growth factor receptor 2.

**Figure 2 jpm-13-00249-f002:**
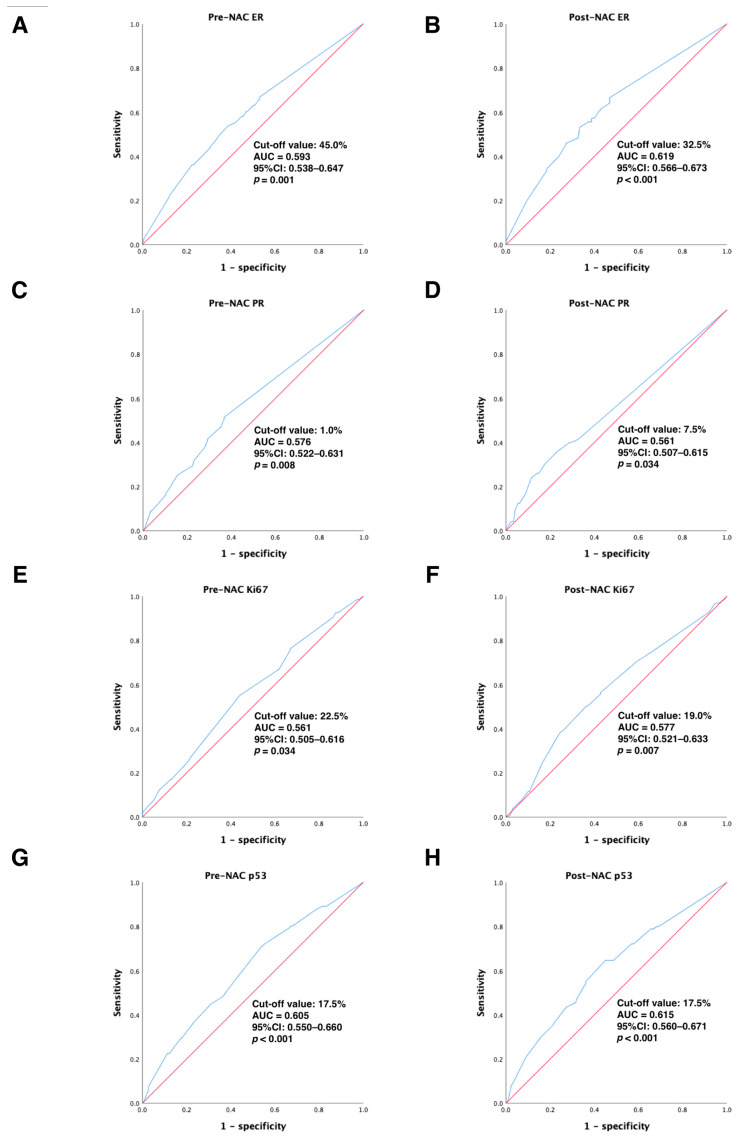
ROC analysis showing the best discriminative value to predict DFS outcomes for (**A**) pre-NAC ER; (**B**) post-NAC ER; (**C**) pre-NAC PR; (**D**) post-NAC PR; (**E**) pre-NAC Ki67; (**F**) post-NAC Ki67; (**G**) pre-NAC p53; (**H**) post-NAC p53 in the primary cohort. Blue line: ROC curve; Red line, reference line. Abbreviations: ROC, receiver operating characteristic; DFS, disease-free survival; AUC, area under the curve; CI, confidence intervals; ER, estrogen receptor; PR, progesterone receptor; pCR, pathologic complete response.

**Figure 3 jpm-13-00249-f003:**
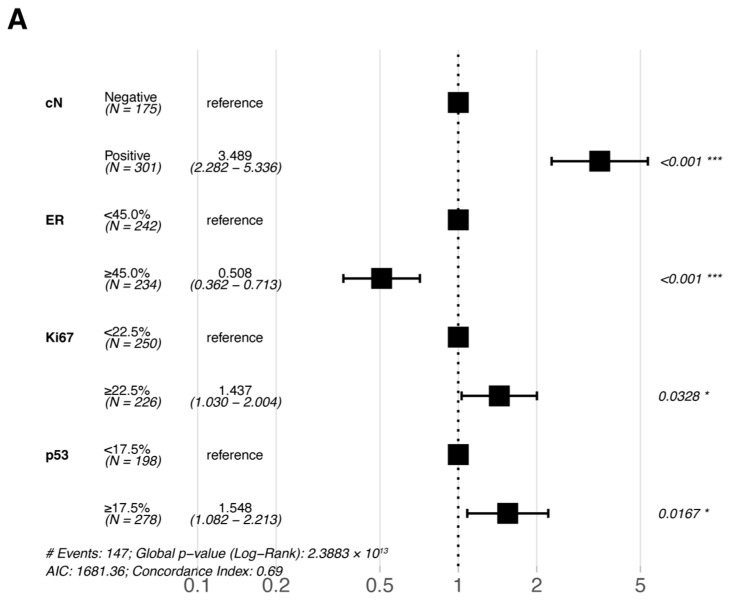
Forest plots of the multivariate analysis of DFS in the primary cohort based on (**A**) pre-NAC core needle biopsy and (**B**) post-NAC surgical specimens. * *p* < 0.05; ** *p* < 0.01; *** *p* < 0.001. Abbreviations: DFS, disease-free survival; pCR, pathologic complete response; NAC, neoadjuvant chemotherapy; cN, clinical nodal; ER, estrogen receptor.

**Figure 4 jpm-13-00249-f004:**
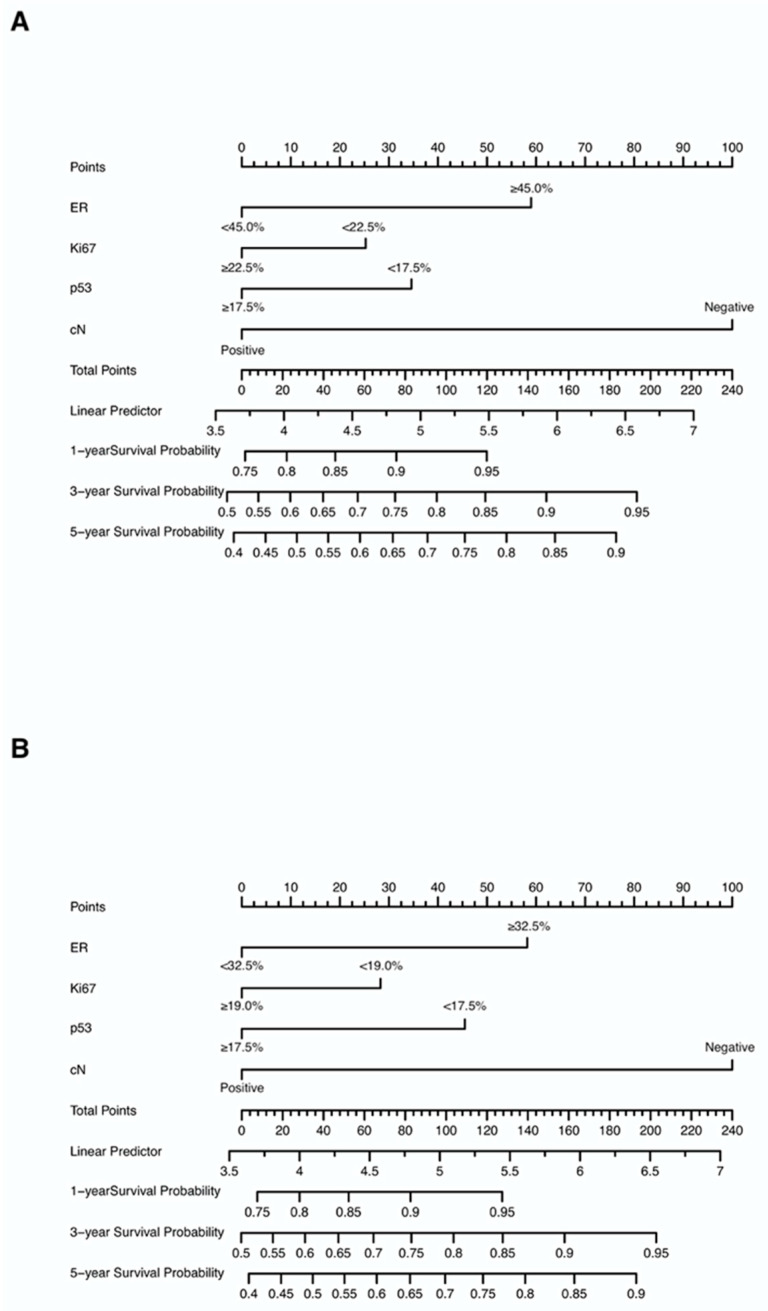
Nomograms predicting the 1-, 3-, and 5-year DFS in non-pCR patients based on (**A**) pre-NAC core needle biopsy and (**B**) post-NAC surgical specimens. Each value of each variable corresponds to some points on the line “Points”; the sum of all points together (Total Points) corresponds to a probability on the lines “1-year Survival Probability”, “3-year Survival Probability”, and “5-year Survival Probability”. Abbreviations: DFS, disease-free survival; pCR, pathologic complete response; NAC, neoadjuvant chemotherapy; ER, estrogen receptor; cN, clinical nodal.

**Figure 5 jpm-13-00249-f005:**
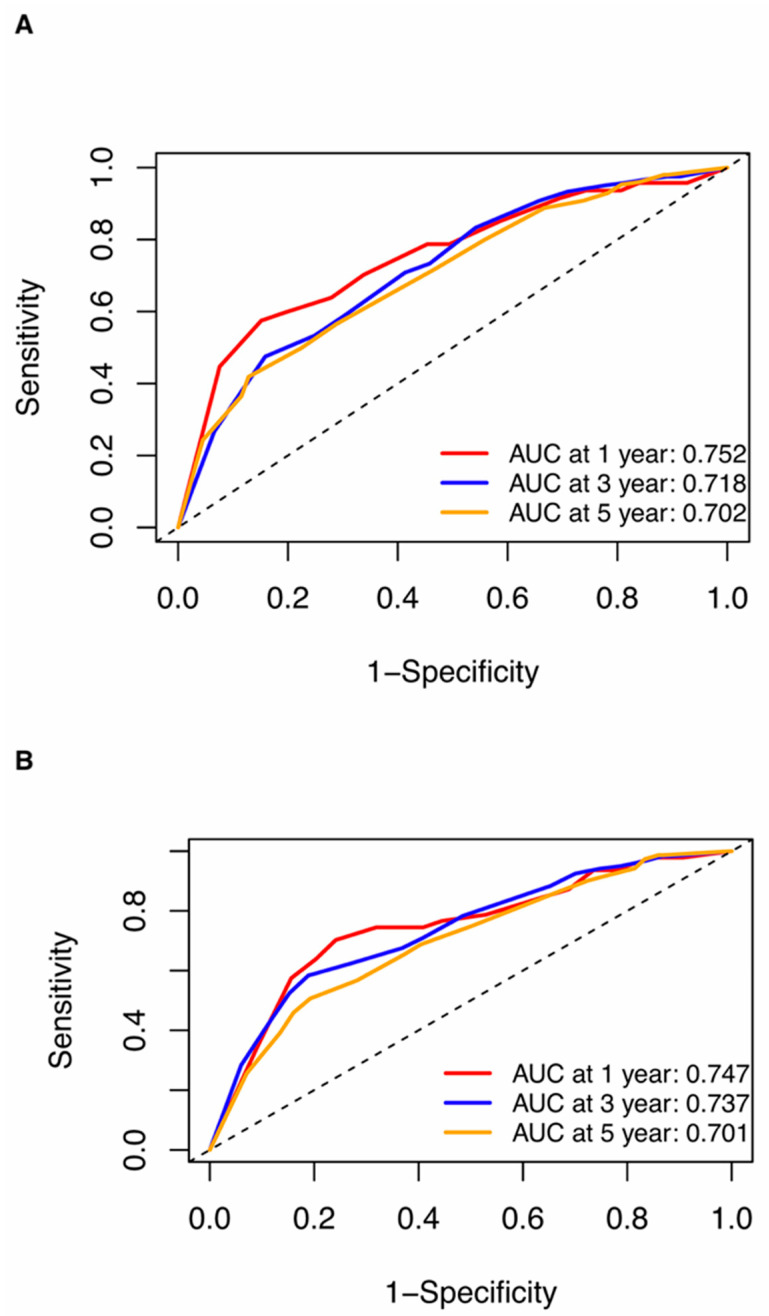
ROC curves for (**A**) the pre-NAC nomogram model and (**B**) the post-NAC nomogram model to predict the probability of 1-, 3-, and 5-year DFS in the primary cohort. Dashed-line: reference line. Abbreviations: ROC, receiver operating characteristic; DFS, disease-free survival; pCR, pathologic complete response; NAC, neoadjuvant chemotherapy; AUC, area under the curve.

**Figure 6 jpm-13-00249-f006:**
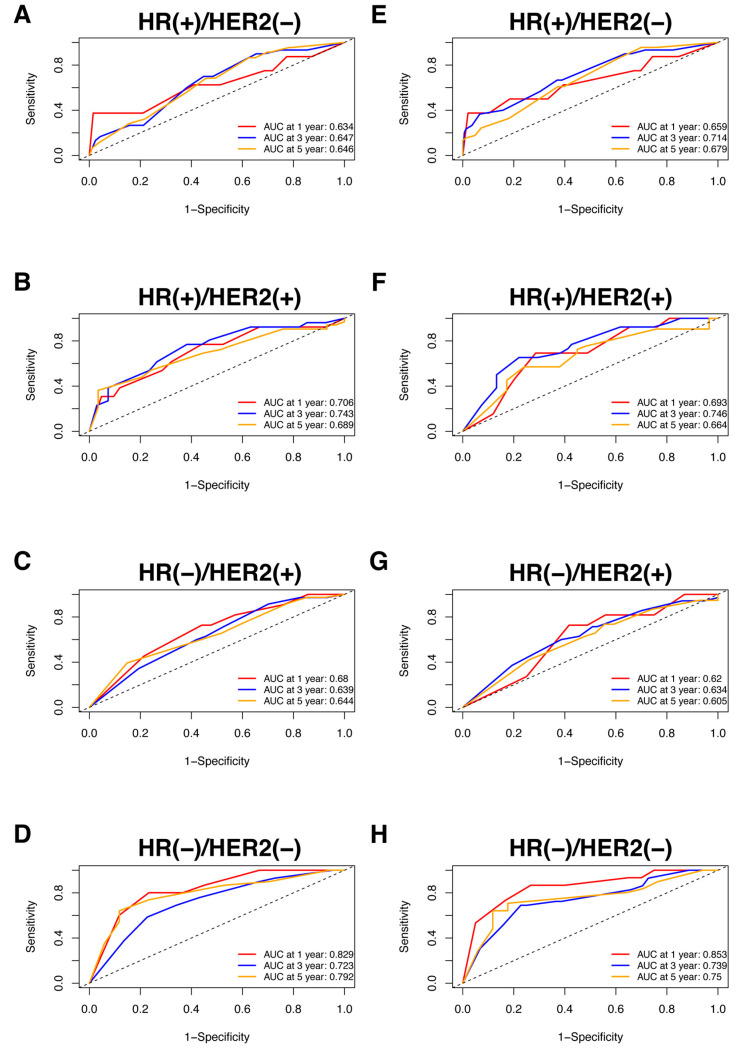
The ROC curve for the prediction performance of the pre-NAC nomogram model in the (**A**) HR(+)/HER2(−), (**B**) HR(+)/HER2(+), (**C**) HR(−)/HER2(+), and (**D**) HR(−)/HER2(−) subgroups and the ROC curve for the prediction performance of the post-NAC nomogram model in the (**E**) HR(+)/HER2(−), (**F**) HR(+)/HER2(+), (**G**) HR(−)/HER2(+), and (**H**) HR(−)/HER2(−) subgroups. Dashed-line: reference line. Abbreviations: ROC, receiver operating characteristic; NAC, neoadjuvant chemotherapy; HR, hormone receptor; HER2, human epidermal growth factor receptor 2; AUC, area under the curve.

**Figure 7 jpm-13-00249-f007:**
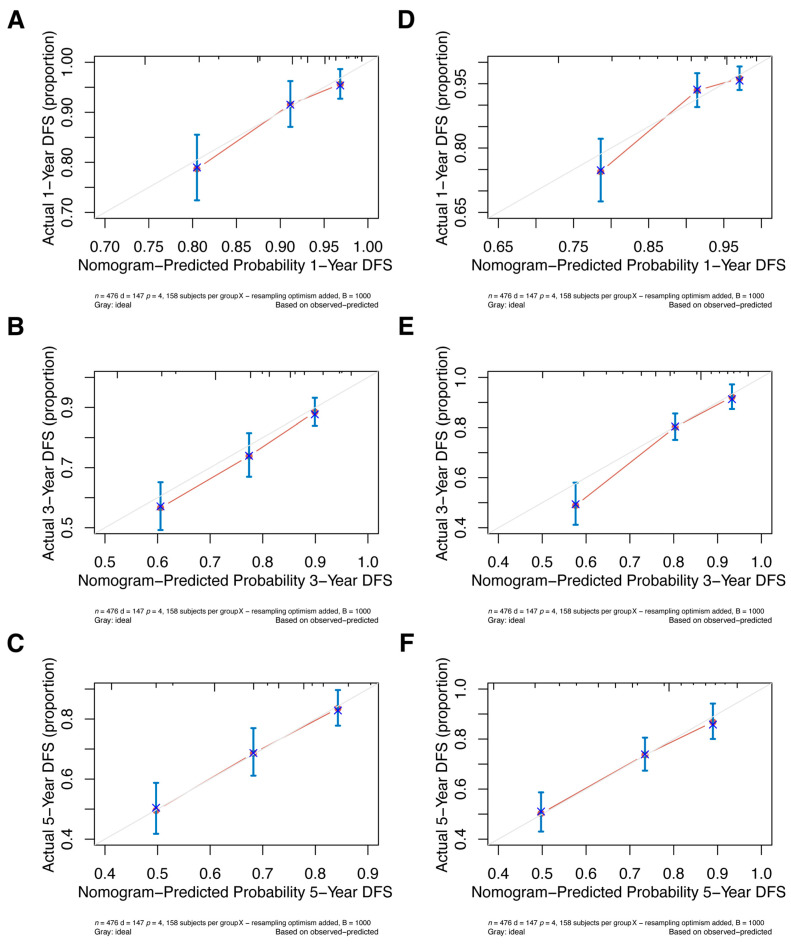
The pre-NAC nomogram model calibration plot for (**A**) 1-, (**B**) 3-, and (**C**) 5-year DFS in the primary cohort; the post-NAC nomogram model calibration plot for (**D**) 1-, (**E**) 3-, and (**F**) 5-year DFS in the primary cohort. Abbreviations: NAC, neoadjuvant chemotherapy; DFS, disease-free survival.

**Figure 8 jpm-13-00249-f008:**
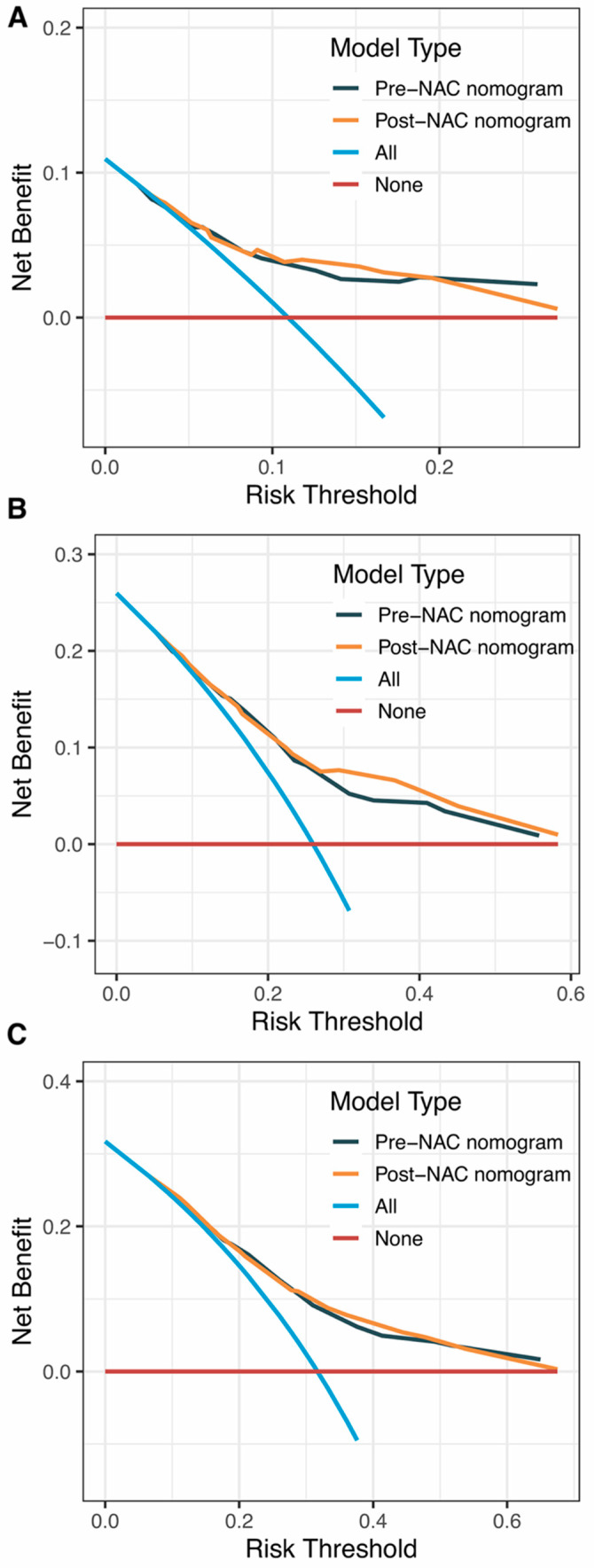
DCA curves for (**A**) predicting 1-year DFS nomograms; (**B**) predicting 3-year DFS nomograms; (**C**) predicting 5-year DFS nomograms. Abbreviations: DCA, decision curve analysis; DFS, disease-free survival; NAC, neoadjuvant chemotherapy.

**Figure 9 jpm-13-00249-f009:**
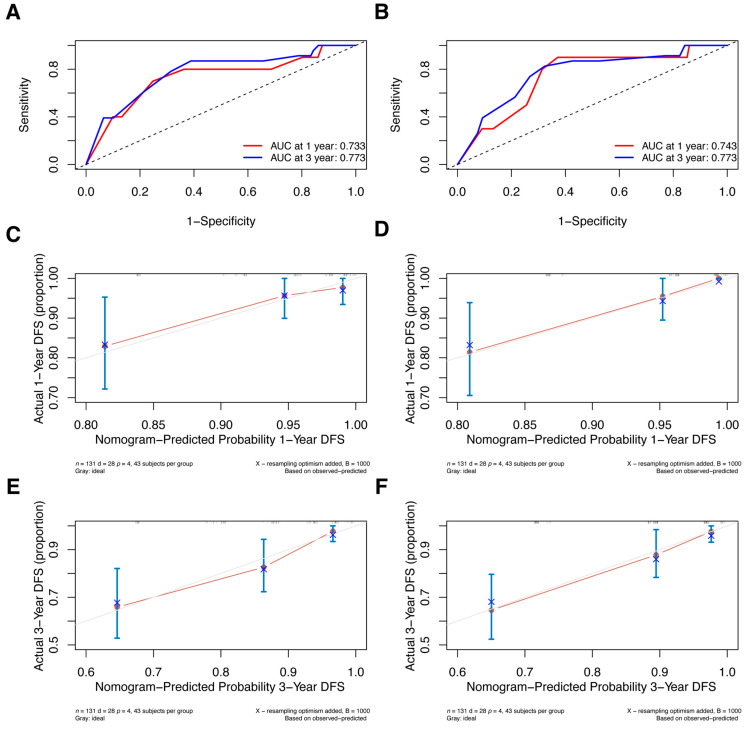
ROC curves for (**A**) the pre-NAC nomogram model and (**B**) the post-NAC nomogram model to predict the probability of 1- and 3-year DFS in the validation cohort. Calibration curves for the pre-NAC nomogram model predicting the probability of (**C**) 1- and (**E**) 3-year DFS in the validation cohort; calibration curves for the post-NAC nomogram model predicting the probability of (**D**) 1- and (**F**) 3-year DFS in the validation cohort. Dashed-line: reference line. Abbreviations: ROC, Receiver operating characteristic; NAC, neoadjuvant chemotherapy; DFS, disease-free survival; AUC, area under the curve.

**Figure 10 jpm-13-00249-f010:**
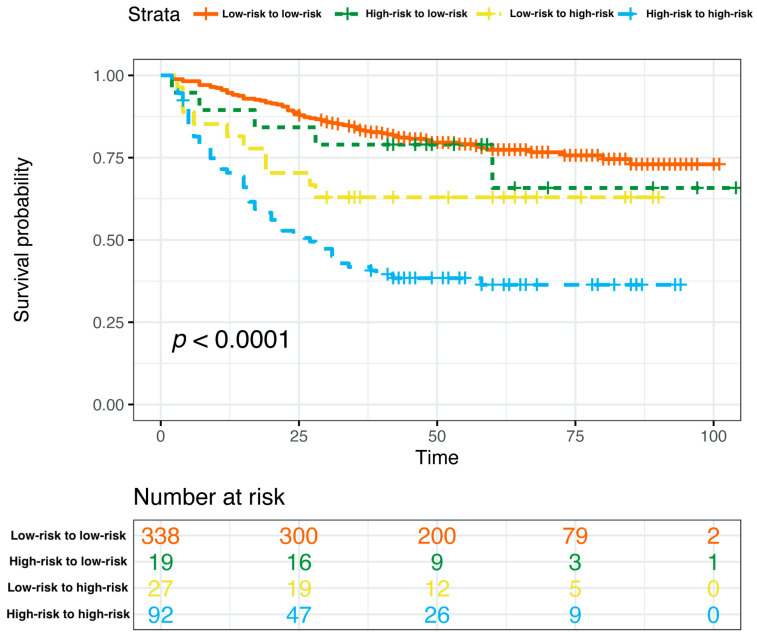
Kaplan–Meier estimates of DFS according to risk category shifts in the primary cohort. Abbreviations: DFS, disease-free survival; pCR, pathologic complete response.

**Table 1 jpm-13-00249-t001:** Baseline clinicopathological characteristics and univariate analysis.

Predictive Factors	Comparison of Predictive Factors between the Two Groups [M (P25, P75)]				
	All Subjects (*n* = 476) *n* (%)	Event Group (*n* = 147) *n* (%)	Non-Event Group (*n* = 329) *n* (%)	Z/x^2^	*p*
Age at diagnosis, y	49.0 (43.0–57.0)	48.0 (42.0–57.0)	49.0 (44.0–57.0)	−1.437	0.151
Menopausal status				0.055	0.841
Pre-menopause	279 (58.6%)	85 (57.8%)	194 (59.0%)		
Post-menopause	197 (41.4%)	62 (42.2%)	135 (41.0%)		
cT				8.758	0.012
cT1	23 (4.8%)	6 (4.1%)	17 (5.2%)		
cT2	333 (70.0%)	91 (61.9%)	242 (73.6%)		
cT3 + cT4	120 (25.2%)	50 (34.0%)	70 (21.3%)		
cN				30.963	<0.001
Negative	175 (36.8%)	27 (18.4%)	148 (45.0%)		
Positive	301 (63.2%)	120 (81.6%)	181 (55.0%)		
Pre-NAC ER status (%)				9.176	0.003
<45.0	242 (50.8%)	90 (61.2%)	152 (46.2%)		
≥45.0	234 (49.2%)	57 (38.8%)	177 (53.8%)		
Post-NAC ER status (%)				16.082	<0.001
<32.5	252 (52.9%)	98 (66.7%)	154 (46.8%)		
≥32.5	224 (47.1%)	49 (33.3%)	175 (53.2%)		
Pre-NAC PR status (%)				8.737	0.004
<1.0	253 (53.2%)	93 (63.3%)	160 (48.6%)		
≥1.0	223 (46.8%)	54 (36.7%)	169 (51.4%)		
Post-NAC PR status (%)				7.583	0.008
<7.5	324 (68.1%)	113 (76.9%)	211 (64.1%)		
≥7.5	152 (31.9%)	34 (23.1%)	118 (35.9%)		
Pre-HER2 status				6.034	0.050
HER2-0	71 (14.9%)	23 (15.6%)	48 (14.6%)		
HER2-low	223 (46.8%)	57 (38.8%)	166 (50.5%)		
HER2-postive	182 (38.2%)	67 (45.6%)	115 (35.0%)		
Post-HER2 status				3.902	0.145
HER2-0	72 (15.1%)	22 (15.0%)	50 (15.2%)		
HER2-low	217 (45.6%)	58 (39.5%)	159 (48.3%)		
HER2-postive	187 (39.3%)	67 (45.6%)	120 (36.5%)		
Pre-NAC Ki67 status (%)				4.956	0.029
<22.5	250 (52.5%)	66 (44.9%)	184 (55.9%)		
≥22.5	226 (47.5%)	81 (55.1%)	145 (44.1%)		
Post-NAC Ki67 status (%)				8.090	0.005
<19.0	282 (59.2%)	73 (49.7%)	209 (63.5%)		
≥19.0	194 (40.8%)	74 (50.3%)	120 (36.5%)		
Pre-NAC p53 status (%)				11.912	0.001
<17.5	198 (41.6%)	44 (29.9%)	154 (46.8%)		
≥17.5	278 (58.4%)	103 (70.1%)	175 (53.2%)		
Post-NAC p53 status (%)				15.685	<0.001
<17.5	233 (48.9%)	52 (35.4%)	181 (55.0%)		
≥17.5	243 (51.1%)	95 (64.6%)	148 (45.0%)		
Chemotherapy cycles				3.005	0.227
3	10 (2.1%)	3 (2.0%)	7 (2.1%)		
4	432 (90.8%)	129 (87.8%)	303 (92.1%)		
5–8	34 (7.1%)	15 (10.2%)	19 (5.8%)		

cT, clinical T staging; cN, clinical nodal status; NAC, neoadjuvant chemotherapy; ER, estrogen receptor; PR, progesterone receptor; HER2, human epidermal growth factor receptor 2.

## Data Availability

The original datasets used and/or analyzed during the present study are available from the corresponding author upon reasonable request.

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
