# Peer review of "Nomograms for Predicting Disease-Free Survival Based on Core Needle Biopsy and Surgical Specimens in Female Breast Cancer Patients with Non-Pathological Complete Response to Neoadjuvant Chemotherapy"

_jpm, 2023, doi:10.3390/jpm13020249_

Round 1
Reviewer 1 Report
Nomograms for predicting disease-free survival based on core needle biopsy and surgical specimens in female breast cancer patients with non-pathological complete response to neoadjuvant chemotherapy
This an interesting and highly valuable premise to investigate
Neoadjuvant chemotherapy (NAC), an effective treatment widely used before surgery, is the current standard of care for locally advanced or high-risk early-stage BC [2], aiming to downstage tumour staging and render inoperable tumours operable. Please define risk stratification in breast cancer. I realise you mention these are the more aggressive subtypes plus the HR+, but this may not mean a lot to a reader who is hoping to learn a little bit about the subtypes by reading this manuscript.
It would also be good if the authors could give more information about pCR about NAC and what this definition is. Usually when a concept like this is introduced, one needs to elaborate on it especially since this is the core of the study.
In lines 50-56, it would be good if this could be moved higher up as to encompass an earlier introduction to molecular subtypes of BC and could include what hormonal profile luminal A/B, HER2+ and TNC have. This would nicely form the opening sentences of the manuscript and also help orient the readers.
Hot research in my opinion could be replaced with trendy research.
The argument about p53 being overlooked in BC is valid and it would be good if the authors could elaborate on the genetic alterations pertinent to each BC subtype.
In addition, studies on the efficacy of NAC for BC have shown that pCR rates are significantly higher in tumours with p53 mutations compared to wild-type tumours [10]. Since you have not defined pCR in a detailed fashion, for a reader not familiar with this terminology/ concept, this sentence will be difficult to analyse. Is having a high pCR favouring survival or the opposite?
Why did the authors only evaluate DFS and not OS as well?
In general, the introduction is abstract and skips explaining some vital information. Understandably, the authors have aimed to write for an expert audience but a little more explanation may also attract readers who are familiar with BC but not these terms specifically or audiences who are familiar with cancer but not BC specifically.
In the methods section, the authors could elaborate on what medication is intended exactly for NAC. This is important since they exclude patients who received certain treatment options in line 87.
The flowchart on page 3 is very useful.
ER, PR, HER2, Ki67, and p53 were measured by IHC. The values of ER, PR, Ki67, and p53 were determined as the percentage of nuclear staining positive cells in cancer. Could the authors attest to the consistency of the processing of this staining?
In lines 104-108, I assume that the authors refer to the staining quantified for the same sample but by two different pathologists.
Lines 112-113: this sentence seems to be a comment that got incorporated into the manuscript:
ER, PR, HER2, Ki67, and p53 were measured by IHC. The values of ER, PR, Ki67, and p53 103 were determined as the percentage of nuclear staining positive cells in cancer. BTW how was this distinction made?
Since there was no way to confirm whether the FISH report was from core biopsy or surgical resection, no distinction was made between HER2 status before and after NAC. Please explain surgically why this was the case and this distinction could not be made.
Finally, pCR and non-pCR have been defined in lines 117-119, this however does not preclude the readers from mentioning them in the introduction.
It would be interesting if the authors could explain why the surgical procedure after NAC would still leave behind come tumour (so a clear margin had not been achieved). Would this be because some tumour tissue was unresectable or was simply missed, please explain the practical examples when non-pCR
DFS events were defined as the first recurrence of the disease locally, regionally, or distantly, diagnosis of contralateral BC or secondary malignancy, or death from any cause. DFS stands for disease-free survival, so basically, the authors mean that DFS ended after the mentioned occurrences.
Since the Bootstrap method has 1,000 internal replicate samples, it is not necessary to divide the data into training and test sets. Even though what the authors say is technically correct, it is still better than a predictive model getting tested on multiple independent datasets. So I would suggest testing your method/ model on at least another dataset and adding this information here.
The risk score distinguishing DFS-event tumours from DFS-free event tumours was estimated using the Maximally Selected Rank Statistics, from which cutpoint values were obtained. Based on cutpoint values, patients were divided into high-risk and low-risk groups. The exact process and definitions for low-risk and how the authors reached this, need to be explained better.
The data relating to figure 2/ table 1: In practical terms what does the cut-off value for each parameter pre and post-NAC mean and what is its significance, please explain. It is not clear either what the implications are. Also, emphasise the point made earlier about a DFS event and the implication of this in lines 175-176 (the occurrence of that event ended DFS etc).
The AUCs in figure 2 are relatively low. Could this be improved?
The authors could release a supplementary file with the anonymous patient characteristics that they used for this study to increase transparency and reproducibility.
The in-figure writing in figure 3 is not legible, please enhance the quality of this figure. It is interesting that cN, Ki67, and p53 increase risk while ER decreases it (under both conditions), why is this and what are the implications?
Most of the analyses of this paper would not be possible in a pCR since the post-NAC surgical sample would not exist.
Fig. 4. The C-index of the pre-NAC and post-NAC nomograms were 0.693 and 0.701, respectively. Please define C-index better in the methods and also here and explain the findings of figure 4 in greater detail. How does a reader interpret figure 4 if this is a 2-dimensional plot?
The AUCs slightly improve in figure 5 and it is encouraging that the actual DFS matches pre-post NAC predictions in figure 6. Please provide a more detailed analysis of the finding of figure 7 and its implications.
On page 12, the risk paragraph needs a better explanation. For example,
Therefore, patients were classified into different risk subgroups, namely low risk (>59.0 in the pre-NAC nomogram and >58.1 in the post-NAC nomogram), and high risk (≤59.0 in the pre-NAC nomogram and ≤58.1 in the post-NAC nomogram)
What is the unit of this categorisation (i.e., what does 58.1 refer to, is this DFS or a particular cut-off)? I assume this is the case since the next sentence mentions this. Also, it would help if the authors could explain in practical terms how this risk transition happens and what are the implications for the patient.
As mentioned earlier, the authors could test, even a fraction of the methods used, on another dataset as validation, if this is meant to be a proof-of-principle method paper.
Reviewer 2 Report
The researchers developed a well-designed study on a clinically relevant subject. However, I raise a point of concern.
As the authors mentioned in lines 281 and 282, another study observed that p53 had no prognostic value in models for mortality in patients with luminal breast cancer (ER/PR+ and HER2-). In fact, p53 mutation is much more common in HER2+ or Triple-negative cancers (approximately 80%) than in luminal ones (12% in luminal A and up to 32% in luminal B) (DOI: 10.1038/nature11412). Since the mutation is much more frequent in cancers with more aggressive biology, the analysis of this factor separately may lead to a spurious or indirect correlation with the analyzed survival. Likewise, the analysis of ER, PR, and Ki-67 correlate with molecular subtypes, the latter being much higher also in HER2+ or Triple-negative subtypes (DOI: 10.1016/j.breast.2015.07.017; 10.1186/ s12885-017-3212-x).
Therefore, the ideal is that the models consider the molecular subtypes (DOI: 10.1093/annonc/mdt303), or that separate nomogram are made for each subtype (or at least for luminal segregated from HER2+/Triple-negatives).
Also, I noticed the lack of histological grade. Although there is some correlation with Ki-67, it measures other important characteristics with high prognostic value.
Finally, in lines 31 and 32, the phrase "Patients in the high-risk to high-risk subgroup have significantly poorer survival rates (P<0.0001)" does not make it clear that there may be a change in risk from pre-NAC to post-NAC. NAC, as it is still the Abstract. Modify so that the reader can understand this right away in the Abstract, without having to read the entire manuscript.
Round 2
Reviewer 1 Report
The authors have addressed my comments.
Reviewer 2 Report
The authors endeavored to remedy the gaps and considerably improved the study.